# The Response and Evaluation of Morphology, Physiology, and Biochemistry Traits in Triploid *Passiflora edulis* Sims ‘Mantianxing’ to Drought Stress

**DOI:** 10.3390/plants13121685

**Published:** 2024-06-18

**Authors:** Xin Su, Zhenxin Yang, Chiyu Zhou, Shili Geng, Shi Chen, Nianhui Cai, Junrong Tang, Lin Chen, Yulan Xu

**Affiliations:** 1Key Laboratory of Forest Resources Conservation and Utilization in the Southwest Mountains of China, Ministry of Education, Southwest Forestry University, Kunming 650224, China; suxin@swfu.edu.cn (X.S.); yzx@swfu.edu.cn (Z.Y.); zhouchiyu@swfu.edu.cn (C.Z.); 1213193664@swfu.edu.cn (S.G.); cainianhui@swfu.edu.cn (N.C.); 2Key Laboratory of National Forestry and Grassland Administration on Biodiversity Conservation in Southwest China, Southwest Forestry University, Kunming 650224, China; chenshi@swfu.edu.cn (S.C.); tjrzy2016@swfu.edu.cn (J.T.); linchen@swfu.edu.cn (L.C.)

**Keywords:** drought stress, *Passiflora edulis*, polyploid, physiology and biochemistry

## Abstract

As one of the most influential environmental factors, drought stress greatly impacts the development and production of plants. Triploid-induced *Passiflora edulis* Sims ‘Mantianxing’ is an important new cultivar for multi-resistance variety selective breeding, which is one of the *P. edulis* breeding essential targets. However, the performance of triploid ‘Mantianxing’ under drought stress is unknown. In order to study the drought resistance of triploid ‘Mantianxing’, our study compared drought-related indicators in diploids and triploids under natural drought experiments, including morphological, physiological, and biochemical characteristics. Results showed that triploid *P. edulis* ‘Mantianxing’ showed variable responses to drought treatment. Compared with diploids, triploids showed higher photosynthesis and chlorophyll fluorescence, osmotic adjustment substances, and antioxidant enzyme activity under drought stress and faster chlorophyll biosynthesis and growth recovery after rewatering. Generally speaking, these results indicate that the drought resistance of triploid *P. edulis* is superior to diploid. This study provides scientific information for breeding stress tolerance variety of *P. edulis* ‘Mantianxing’ new cultivar.

## 1. Introduction

*Passiflora edulis* Sims ‘Mantianxing’ (2n = 18) is one of the main passion fruit cultivars cultivated in tropical and subtropical regions [1]. It is widely used in food processing due to its unique flavor. In recent years, it attracted much attention in the medical field and pharmaceutical industry [2,3] for containing various active phytochemical constituents, such as tannins, coumarin alkaloids, flavonoids, tyrosine, glycine, phenols, and polysaccharides [4,5].

Polyploid refers to a creature that owns three or more sets of complete chromosomes [6], which is considered the dominant feature of plant diversity [7]. Polyploidization universally occurs in nature, and numerous studies have identified that polyploid plants have a more abundant variation, whether in morphology, physiology, or biochemistry, than normal diploids [8,9,10]. For example, the triploid *Eucommia* has a more abundant variation amplitude and shows advantages in phenotype, photosynthesis, and secondary metabolites than diploid [11]. Similarly, triploid tea *Camellia sinensis* shows a significantly different morphological structure; its leaf area is 59.81% larger than that of diploid tea, and its photosynthetic capacity is greater. These variations contributed to the breeding of high-quality and high-yield tea varieties [12]. Compared with triploid, tetraploid genotypes in C_4_ grass *Miscanthus x giganteus* could produce economically friendly leaf morphological changes, such as thicker leaf and greater leaf mass per area, which may have a practical contribution to improve photosynthetic efficiency and crop productivity [13]. Additionally, polyploidization is beneficial for plants to evolve a more favorable survival strategy. The relevant study suggested that the changes in genome duplication improve physiology and biochemistry traits in tetraploid *Jasione maritima*, which makes for its environmental adaptations and tolerances under water deficit [14]. Moreover, hexaploidy *Achillea borealis* (Asteraceae) shows a fivefold fitness advantage and higher survival rate over tetraploids in dune habitats, which means better ecological adaptation [15]. Thus, in recent years, researchers have widely used polyploids to study biotic and abiotic resistance in order to improve the quality and yield of target plants under stress. However, spontaneous polyploid *P. edulis* is still infrequent, and most polyploids were obtained by artificial inducers, and knowledge about polyploid *P. edulis* remains scarce, especially the resistance performance of polyploid *P. edulis*.

According to relevant studies, more and more frequent and prolonged drought stress caused by climate change, as the most severe natural hazard and abiotic stress, limits agricultural production and crop yield to a great extent [16]. Drought stress could change the growth and development of plants, such as reducing leaf length, area, fresh weight, water movement [17], photosynthesis capacity [18], variation in biomass distribution [19] and root morphology [20], and reduction in chlorophyll content [21]. Additionally, water deficits caused by heat stress exacerbate the reproductive process of a variety of cereals and legumes, resulting in crop failure [22]. In order to protect from water deficits and heat waves, different plants have adapted various response mechanisms. For instance, improving cellular water retention via osmotic adjustment [23], keeping drought resistance by sugar signaling and transporting [24], soluble protein accumulation [25], and improving photosynthetic [26] as well as antioxidant activity to defend against oxidative stress [27]. Meanwhile, studies on combining drought resistance in various plants have found that plants with higher ploidy levels showed better performance under stress [28,29,30]. However, whether the theory holds true for polyploid *P. edulis* remains to be seen.

At present, research about polyploid *P. edulis* ‘Mantianxing’ has mainly focused on a comparison of differences in morphology, photosynthetic characteristics, and chlorophyll and chlorophyll fluorescence parameters between diploid and triploid *P. edulis* ‘Mantianxing’, finding that triploid differs from diploid in phenotype (plant height, leaf size, etc.) and had a higher photosynthetic capacity. Moreover, triploids had higher photosynthetic pigment and fluorescence yield, better light quanta energy, and a higher photosynthetic carbon metabolism rate [31,32]. These findings provided references and facilitated genetic improvement for further breeding programs of *P. edulis* selection. However, whether triploid and diploid also perform differently under drought stress is unclear. Based on the above findings of triploid and diploid ‘Mantianxing’, we hypothesized that the triploid and diploid ‘Mantianxing’ probably differ in morphological, physiological, and biochemical responses under drought stress. Therefore, triploid and diploid ‘Mantianxing’ were used as material to evaluate their differences in physiology and biochemistry characteristics under drought stress in the present study to provide more scientific information for the breeding project of drought-resistant polyploid *P. edulis* ‘Mantuianxing’ cultivars and promote further development of *P. edulis*-related industries.

## 2. Materials and Methods

### 2.1. Plant Material

The current study involved determining and analyzing physiology and biochemistry traits in triploid and diploid ‘Mantianxing’ under drought stress to evaluate their resistance. The material of definite diploid (2n = 2x = 18) was obtained by sterile embryo germination. Triploid aseptic seedlings (2n = 3x = 27), including three plant lines named T4, T10, and T17, respectively, were successfully induced by endosperm culture [32].

### 2.2. Drought Stress Experiment

On 8 January 2022, a rooting culture of diploid and triploid ‘Mantianxing’ sterile seedlings was conducted. One plant per culture flask was placed in tissue culture room at Southwest Forestry University with a temperature of (25 ± 2) °C and an average of 12 h/d of light/darkness; the light intensity was 1000~1800 lux. After 60 days, we selected 70 sterile seedlings per plant line with stable growth conditions to soak in 0.1% carbendazim for 10 min and then transferred to 8-liter plastic basin with sterilized soil matrix containing red soil, humus soil, and perlite (1:2:1). All seedlings were cultured in the Nursery of Southwest Forestry University (25°04′0″ N, 102°45′41″ E, Kunming, Yunnan, China). In order to reduce the impact of underground vapor on the experiment, the plastic film was mulched on the ground. On 18 July 2023, 60 one-year-old seedlings with stable growth conditions were randomly selected to implement a natural drought experiment. The natural drought stress experiment was set up with four stress treatment times (drought 0-day, drought 7 days, drought 14 days, drought 21 days) and rewatering 5 days (irrigating 3 L water at 8:00 and 18:00 every day). Three seedlings were placed in each plant line per treatment, with one plant as an independent replication and three biological replications. Materials of four plant lines of drought 0-day, 7 days, 14 days, 21 days, and rewatering 5 days (R5) after drought for 21 days were respectively collected at 8:30 am for further measurements.

Plant physiology and biochemistry indicators were measured, including leaf morphology indicators, root length, biomass, chlorophyll content, soluble protein and soluble sugar, enzyme activity, including superoxide (SOD), peroxidase (POD), catalase (CAT), malonaldehyde (MDA), and osmotic regulator proline (Pro), and photosynthetic characteristics.

### 2.3. Determination of Leaf Morphology Indicators, Growth Yield, and Biomass 

Morphology indicators, including leaf length and leaf width, were measured with tape measure and vernier caliper, respectively. Measured leaves were the 3rd to 5th leaves from the top of the plant [33]. Twelve plants were measured in each plant line. Root length was recorded and analyzed by Plant Image Analyzer (WanShen LA-S series; Phantom 9900XL Plus). The biomass measurement mainly included the above-ground biomass of stem and leaf and the underground biomass of root, with one plant as a replication and three biological replications. The materials were washed and drained gently, and then the seedlings were divided into three parts (root, stem, and leaf) to measure and record the fresh weight of each part. (The figure is accurate to three decimal places.)

### 2.4. Physiological Index Measurements

Before the measurement of growth yield and biomass, the physiological indicators were recorded. Leaf materials selected the third to fifth leaves on the top of the plant, one leaf as a replication, and three biological replications.

We measured chlorophyll fluorescence parameters at 8:30 am using plant efficiency analyzer (Handy PEA). Parameters included the following: minimum fluorescence yield in the absence of photosynthetic light (*F*_0_), maximum fluorescence yield in the absence of photosynthetic light (*F*_m_), variable fluorescence (*F*_v_), the maximum quantum yield of PS II (*F*_v_/*F*_m_), electron transfer rate (*F*_m_/*F*_0_), the ratio of photochemical to nonphotochemical processes (*F*_v_/*F*_0_), light energy absorbed by a unit reaction center (ABS/RC), dissipation at the level of the antenna chlorophylls (DIo/RC), the specific energy fluxes (per reaction center, RC) for trapping (TRo/RC), energy captured per unit reaction center for electron transfer (ETo/RC), probability that an absorbed photon will move an electron into the electronic transport chain (ΦEo), the PS Ⅱ maximum photochemical efficiency (ΦPo).

LI-6800 Convenient Photosynth Meter was used to determine the diurnal changes in photosynthetic characteristics of ‘Mantianxing’ at 9:00 am, and indicators, including the net photosynthetic rate (*P*_n_), transpiration rate (*T*_r_), stomatal conductance (*G*_s_), and intercellular carbon dioxide concentration (*C*_i_) were measured.

According to Arnon’s measurement [34], the chlorophyll *a*, chlorophyll *b* content, total chlorophyll content, and carotenoid content were extracted by acetone. The corresponding values of wavelength 663 nm, 645 nm, and 652 nm were respectively recorded by spectrophotometer, and all indicators were calculated according to the following formulas.

Soluble protein content was measured by adapting the Coomassie Brilliant Blue G-250 method [35]. The ethanol method [36] determined the soluble sugar content.

### 2.5. Determination of Antioxidant Enzymes

Pro, MDA, and the antioxidant enzyme activities of CAT, SOD, and POD were measured using commercial kits, which were bought from Suzhou Michy Biomedical Technology Co., Ltd. (Suzhou, China). The specific operation steps were strictly implemented in accordance with the instructions provided by the company free of charge. All the parameters were measured using the same enzyme calibration.

### 2.6. Statistical Analysis

Excel (2019) was used to record and sort data; single factor analysis of variance was conducted using IBM SPSS statistics 22.0. Duncan’s method was used to test the significance. All calculations and drawings are represented by means ± standard errors. All graphs were completed via Origin 2021 and Adobe Illustrator 2021.

## 3. Results

### 3.1. The Effect of Drought Stress on the Morphology of Diploid and Triploid P. edulis ‘Mantianxing’

According to the results, drought stress changed the morphological traits of different ploidy plants to different degrees. From Figure 1, the morphology of both ploidy levels showed a good growth state, and the triploid leaves were greener in comparison with diploid before drought stress. With a decrease in the water content in plants, all plants showed a withered phenotype gradually. With the severity of drought, the leaves significantly withered and fell, and the stem dried up. It was worth mentioning that the triploid plants mainly supplied nutrients and water to the growth of lateral shoots and leaves under drought stress, especially the T4 and T17 plants, which put out more new leaves in 21 days of drought, while the new shoots and leaves of diploid spouted after rewatering. Moreover, all new shoots in diploid and triploid showed rapid growth after rewatering. Additionally, it was found that the new shoots and leaves of T17 and T4 showed better growth than T10 and diploid during the later drought periods of 14 days to 21 days. A significantly gradual decrease in leaf length and leaf width induced by drought is shown in Table 1 (*p* < 0.05), but after rewatering 5 days, the size of the leaves of both diploid and triploid already recovered to before stress.

### 3.2. The Response of Organ Biomass in ‘Mantianxing’ to Drought Stress

On the whole, dehydration caused by drought stress brought about a significant decrease in biomass in both diploid and triploid ‘Mantianxing’. From Figure 2, the above and underground biomass of diploid and triploid showed a downward trend with prolonged stress time, and the minimum value occurred in the treatment of drought 21 days, then slightly recovered after the water recovered. The response of biomass allocation of different organs to drought stress was observed, and a significant variation in biomass allocation was found among four ploidy plants (*p* < 0.05). In the measurement of diploid under five treatments, the changes were mainly shown in the stem biomass and root. A slight fluctuation was observed in the biomass of the leaves, which showed a trend of increasing first and then decreasing with drought degree. The stem biomass increased under 14 days of drought treatment, then decreased to the minimum allocation percentage of 48.07% after 21 days of drought but returned to the maximum of 57.71% in R5 treatment. The underground root biomass of diploid decreased to a minimum of 22.04% after 14 days of drought stress before recovering to the maximum allocation of 34.04% after 21 days of drought. The biomass of the stem and leaf of the T4 line increased with stress time, then decreased to the smallest allocation percentage of 44.71% and 21.30% in the R5 treatment, respectively. On the contrary, a declining tendency appeared in the root biomass with stress severity, and the largest percentage of 33.99% appeared in the treatment of rewatering 5 days. The biomass allocation percentage of leaf in T10 showed a rising trend to a maximum percentage of 26.20% with drought stress and significantly fell back to 21.79% in R5. The biomass allocation percentage of root maintained a continuously growing tendency with drought degree, while stem presented a decreased trend to a minimum of 43.91% after 21 days of drought. The biomass of leaves in T17 changed slightly; with worsening drought, the biomass percentage of stems increased to a maximum of 68.49%, while that of roots decreased to a minimum of 14.37% after 21 days of treatment. In R5, the biomass allocation percentage of stem significantly decreased to 62.69% while that of root increased to 21.27% (Figure 3). 

### 3.3. Chlorophyll Content, Soluble Protein, and Soluble Sugar 

As can be seen from Figure 4, significant effects of drought stress on the chlorophyll content and carotenoids in different ploidy plants were observed. The chlorophyll *a*, chlorophyll *b*, total chlorophyll, and carotenoids of both diploid and triploid were significantly reduced by drought stress; however, triploid plants still maintained a significantly higher content than diploid in the whole process, especially the chlorophyll content of T4 and T17. Additionally, by the fifth day after rewatering (R5), compared with drought 21 days, the triploid T4, T10, and T17 had recovered 56.89%, 53.37%, and 58.72% of their chlorophyll *a*, respectively, while diploid had recovered 30.82%. The effect of drought stress on chlorophyll *b* was the same as that of chlorophyll *a*; compared with the chlorophyll *b* content after 21 days of drought, triploid T10 and T17 had recovered 98.98%, and 85.78% after 5 days of rewatering, which were higher than the 70.09% of diploid. Similarly, the carotenoids in triploid T4, T10, and T17 had recovered 119.31%, 127.01%, and 84.09% respectively, after rewatering 5 days, while diploid recovered 38.21%. The content of soluble sugar showed a similar changing trend as the above indicators, while the soluble protein content increased first and then decreased significantly with the degree of drought stress. After drought for 21 days, the soluble protein content in T4, T10, and T17 was 64.38%, 61.27%, and 81.24% higher than diploid, respectively. The protein content of diploid after drought stress for 21 days decreased by 63.58% compared with before stress, presenting a greater decline. Similarly, the soluble sugar content in T4, T10, and T17 decreased by 49.83%, 52.00%, and 54.42%, respectively, while diploid decreased by 69.12% (Figure 5). During the whole period of drought, triploid plants showed a higher soluble protein and soluble sugar than diploid. 

### 3.4. Physiological Traits in Diploid and Triploid ‘Mantianxing’ under Drought Stress

From Figure 6, it is observed that the photosynthesis of both diploid and triploid is significantly affected by water inhibition to various degrees. All of the trends of *P*_n_, *T*_r_, *G*_s_, and *C*_i_ in the two ploidy plants significantly decreased with the aggravation of drought and recovered after rewatering, showing a V-shaped curve on the whole. It is worth noting that the values of photosynthesis indicators in the three triploid plants were higher than those in the diploid ones. At the highest drought severity, all photosynthesis indicators significantly decreased in four plant lines (*p* < 0.05). The *P*_n_ of T4, T10, and T17 after 21 days of drought stress were 38.28%, 27.86%, and 37.03% higher than that of diploid, respectively. Meanwhile, the *P*_n_ of T4, T10, and T17 were 29.14%, 30.46%, and 28.71%, significantly lower than the value before stress (0 days), respectively, while the value of diploid decreased by 41.31%, suggesting diploid had a greater decline. Additionally, *T*_r_, *G*_s_, and *C*_i_ presented change trends similar to *P*_n_. The *T*_r_ and *G*_s_ in triploid plants were all higher than diploid. Comparatively speaking, the fluctuation of *C*_i_ was gentler, especially T17. Overall, triploid showed better photosynthetic parameters. 

The effects of drought stress on chlorophyll fluorescence of ‘Mantianxing’ leaves are shown in the figure below. Fluorescence parameters *F*_0_, *F*_m_, and *F*_v_ significantly decreased with the intensity of drought stress and recovered after rewatering. Compared with before stress, the *F*_0_ of diploid significantly decreased by 58.41%, and T4, T10, and T17 decreased by 49.76%, 45.69%, and 48.28%, respectively. The decline amplitudes of *F*_m_ in diploid, T4, T10, and T17 were 66.09%, 55.25%, 52.57%, and 54.32%, respectively, while those of *F*_v_ in four plant lines were 69.14%, 59.16%, 57.87%, and 58.78% in turn. A significant difference was observed in the comparison of diploid and triploid (*p* < 0.05). From Figure 6, we found that the fluctuation of *F*_m_/*F*_o_, *F*_v_/*F*_o,_ and *F*_v_/*F*_m_ in diploid and triploid were similar, and a slight decline was observed after 21 days of drought stress (Figure 7). After rewatering for 5 days, all parameters returned to normal.

### 3.5. Energy Flow and Distribution in PS Ⅱ Reaction Center of Diploid and Triploid 

According to the results, ABS/RC, DIo/RC, ETo/RC, and TRo/RC all showed a downtrend, which significantly decreased to the lowest after 21 days of drought stress and then recovered to normal levels in both diploid and triploid after rewatering. Compared with triploids, significantly higher fluorescence parameters were observed in diploids during the stress experiment. On the contrary, the ΦEo and ΦPo in diploid and triploid showed an increasing trend that reached a maximum after 21 days of drought and fell back to normal at the treatment of rewatering 5 days. In comparison, the ΦPo of triploids was significantly higher than that of diploids under treatments of drought stress. However, after rewatering, a significant difference in ΦEo only appeared in the T4 plant, and no significant difference in ΦPo was found in the four plant lines, although the ΦPo of triploid plants was relatively higher (Figure 8). 

### 3.6. The Antioxidant Enzyme and MDA, Pro in Diploid and Triploid ‘Mantianxing’ under Drought Stress

Antioxidant enzymes, as the most sensitive indicator of plants for detecting environmental stress, could be used to reflect the oxidation resistance of plants under stress. From Figure 9, the oxidation resistance of ‘Mantianxing’ increased significantly with the deepening of drought stress. Triploid plants maintained a significantly higher SOD, POD, CAT, and Pro than diploids during the whole experiment. According to our results, the activity of SOD and CAT in ‘Mantianxing’ increased first and then decreased and recovered after rewatering, while POD, Pro, and MDA increased with prolonged drought stress, then declined after rewatering. The SOD activity of diploid and triploid plants reached the highest value after 7 days of drought stress and then significantly decreased to a minimum value after 21 days of drought treatment. The SOD activities of T4, T10, and T17 were 10.68%, 8.58%, and 13.51% significantly higher than diploid after 7 days of treatment, and were 13.85%, 10.42%, and 12.59% significantly higher than diploid after 21 days of treatment, respectively. The activity of CAT in ‘Mantianxing’ reached a peak after 14 days of drought stress, and T4, T10, and T17 were 18.98%, 16.83%, and 23.49% significantly higher than diploid. Additionally, the POD and Pro of both diploid and triploid reached their maximum after 21 days of drought. Meanwhile, compared with 0-day, the POD of triploid T4, T10, and T17 after 21 days of treatment increased by 39.64%, 51.18%, and 42.69%, respectively, while diploid increased by 54.51%. Similarly, the Pro of T4, T10, and T17 after 21 days of treatment increased by 96.00%, 86.63%, and 96.46%, respectively, and diploid increased by 161.02%. It indicated that triploid had a significantly smaller variation amplitude of POD and Pro than diploid (*p* < 0.05).

The changes in MDA content in diploid and triploid ‘Mantianxing’ leaves under drought stress are shown in Figure 7. Under normal growth conditions, the MDA in diploid only was significantly higher than triploid T10. However, the MDA of diploid and triploid showed an increasing trend as drought conditions worsened and peaked after 21 days of drought, and the MDAs of diploid were 10.22%, 12.21%, and 7.22% higher than T4, T10, and T17, respectively (Figure 9).

### 3.7. The Correlation Analysis of Indicators in Diploid and Triploid ‘Mantianxing’

The correlogram demonstrates the correlation relationships of different indicators of diploid and triploid, respectively, under drought stress. From Figure 10 and Figure 11, the root fresh weight, stem fresh weight, and leaf fresh weight positively correlated with *P*_n_, *T*_r_, *G*_s,_ and *C*_i_. All the antioxidant enzyme activity of both diploid and triploid were extremely positively correlated with the drought stress severity and negatively correlated with organ biomass allocation, soluble sugar, soluble protein, chlorophyll content, and carotenoids, except SOD. Other physiological traits, such as soluble sugar, chlorophyll content, and photosynthetic parameters, were negatively correlated with drought treatment. The soluble sugar, soluble protein, chlorophyll *a*, chlorophyll *b*, and total chlorophyll content, and carotenoid were significantly positively correlated with photosynthetic parameters and chlorophyll fluorescence. 

## 4. Discussion

### 4.1. Effect of Drought Stress on Morphology and Biomass Allocation of Triploid P. edulis ‘Mantianxing’

As revealed by this study, the morphologies of diploid and triploid were gradually impaired with an increase in drought severity. During the course of the drought period, the leaf gradually wilted and shriveled, even falling off, especially under the treatment of 14 days of drought and 21 days of drought, demonstrating that drought stress had a serious impact on the normal growth of the plants [37]. According to a previous study, an imbalance in water levels in a plant and a decrease in cellular water potential and expansion pressure due to water deficiency may result in leaf wilting and abscission [38]. Additionally, after rewatering, new leaves rapidly sprouted in both diploid and triploid lines, supporting the findings that *Quercus acutissima* seedlings would stimulate the shedding of unhealthy and senescent leaves and sprout new leaves to reduce transpiration and maintain water content so as to alleviate drought damage [39]. What is more, from the correlation analysis, there was a positive correlation relationship between the leaf fresh weight and *P*_n_ leaf fresh weight, and *P*_n_ decreased under drought conditions, demonstrating that *P. edulis* ‘Mantianxing’ may adopt escape strategies mainly with premature senescence and leaf reduction to deal with drought stress. However, premature senescence and leaf reduction degraded the chloroplasts and destroyed stromal enzymes, finally reducing photosynthesis [40]. Two ploidy-level plants showed a difference in terms of organ biomass allocation. Among the four plant lines, the whole plant biomass showed a downtrend. Moreover, the organ biomass allocation of diploid and triploid *P. edulis* positively correlated with antioxidant enzymes SOD. In the results on antioxidant enzymes, the SOD content increased first, then decreased. As a previous study suggested, the stressful environment and detoxifying O_2_•-radicals of SOD caused the production and accumulation of H_2_O_2_, which could result in a biomass decline in plant tissues [41]. The root length of all plants increased after 7 days of drought treatment, and the root biomass allocation percentage of diploid and T10 showed an increasing trend under stress. It indicated that ‘Mantianxing’ was tempted to obtain more nutrition from the underground environment under drought to adjust to the stress. This was consistent with previous research into summer maize, which would give priority to dividing water to roots to form larger roots when water was deficient; as a result, the root biomass increased and then transported as much nutrients to the ground as possible [42], but also corresponded with previous findings that evergreens and *Molinia* distributed relatively more biomass to the roots at low nutrient supply, thus increasing their competitive ability for belowground resources [43]. Additionally, adjustment of the above-ground biomass allocation was one of the important components of the adaptive strategies for plant growth [44]. In our findings, stem biomass in two ploidy levels always kept the largest allocation percentage, and that of T4 and T17 presented an increasing trend under stress, suggesting that stems were the important water storage organs of plants. According to a study about *Salsola collina* in response to soil nutrients, plants tended to allocate more biomass to the stem than to the development of reproductive organs when the soil had a lower water content [45]. In summary, plants would adjust their ecological strategies, including morphological traits and biomass allocation patterns, which are affected by environmental factors and ontogeny, so as to have a better adaptation under various growth conditions [46]. Additionally, triploid showed a faster growth recovery after rewatering; this may be attributed to the comprehensive adjusting of morphology, physiology, biochemistry, and drought resistance mechanism in polyploid plants. For instance, more stable physiology and biochemistry traits and specialized microphenotypic structures, such as a thicker epidermis and palisade tissue, caused by chromosome doubling might lead to greater capacity in coping with drought stress [47]. Nucleotypic effects of genome size and increased reproductive flexibility contribute to the polyploid advantage in plants in stressful environments [48].

### 4.2. Variations in Chlorophyll Content and Photosynthesis in Triploid ‘Mantianxing’ under Drought

Under drought, the chlorophyll content of both diploid and triploid ‘Mantianxing’ decreased; according to a previous study, this may be related to chlorophyllase, which plays a vital role in the process of chlorophyll degradation [49,50]. Meanwhile, the leaf morphology size fluctuated unstably due to water inhibition. Leaf size and shape affect light capture, and chlorophyll content is an important factor in photosynthesis [51]. It may explain the decline in *P*_n_ in the two ploidy levels to some extent. What is more, the changing trend in *P*_n_ was similar to that of *G*_s_, suggesting stomatal adjustment may contribute to it. Drought triggered stomatal closure so as to prevent water losses and thus mitigate the negative effects of drought [52]. Stomatal closure and reduced mesophyll conductance affected CO_2_ diffusion from the atmosphere to the site of carboxylation, consequently decreasing photosynthesis under most water stress conditions [53]. During the period of drought, triploid lines always kept a higher chlorophyll content and photosynthesis parameters than diploid, indicating the significant impact of ploidy levels on the physiology characteristics of the plants. As stated in Yang’s research, triploid ‘Mantianxing’ indeed had a higher level of chlorophyll content and stronger photosynthetic capacity than diploid [32], which also may explain the greener leaves of triploid. This improvement may be related to the regulation of the chlorophyll synthesis gene, which is the outcome of whole genome duplication [54]. A variety of studies have shown that an increase in gene dosage enhances the photosynthesis of polyploids under stress [55,56]. 

### 4.3. Chlorophyll Fluorescence and Photosynthesis System under Drought 

Chlorophyll fluorescence is closely correlated with photosynthesis capacity and the physiological state of plants and is used to detect plant physiology and study the impact of stress on photochemistry [57]. The changes in photochemical properties in photosystem II (PS II) can reflect the degree of damage to plant photosynthetic organs that are highly sensitive to soil moisture [58]. Under drought, the chlorophyll fluorescence parameters in diploid and triploid ‘Mantianxing’ were negatively correlated with drought treatment and positively correlated with *P*_n_, *G*_s,_ and chlorophyll content, suggesting that the changes in chlorophyll fluorescence in diploid and triploid plants decreased with prolonged drought stress and with declines in *P*_n_, *G*_s_, and chlorophyll content. It suggests that the decline in photosynthesis activity may be the comprehensive outcome of decreased chlorophyll content, stomatal inhibition, and non-stomatal factors under drought stress [59]. Research into the effect of drought stress on winter wheat has shown similar results: chlorophyll fluorescence parameters decreased under drought [60]. What is more, the changes in energy flow and distribution in PS Ⅱ reaction centers, such as ABS/RC, DIo/RC, ETo/RC, and TRo/RC, indicated that drought stress indeed damaged the photosynthetic system. It is worth noting that, in this study, triploid showed a higher chlorophyll fluorescence than diploid, which is supported by a previous study on triploid ‘Mantianxing’ [31]. Higher chlorophyll fluorescence was widely observed in another polyploid plant [61], even under stress conditions [62,63].

### 4.4. Osmotic Adjustment Substances of Triploid ‘Mantianxing’ under Drought Stress

Soluble protein and soluble sugar are important substances that contribute to plant growth and development. The sugar storage capacity of higher plants is of crucial importance for plant development and adaptation to environmental signals [64]. Meanwhile, as one of the products of photosynthesis, our findings were the same as a previous study that found a positive relationship between sugar content and photosynthesis [65]. In research into *Oryza sativa* L., the authors concluded that photosynthesis significantly impacted soluble sugar content under drought stress [66]. In this study, the downtrend in the soluble sugar content of both diploid and triploid ‘Mantianxing’ may be attributed to the limitation of photosynthesis under drought. Additionally, the changes in soluble sugar and Pro were considered the main osmotic substances of plants under drought stress. We found that Pro sharply increased with prolonged drought, suggesting that ‘Mantianxing’ may change its Pro so as to regulate osmotic adjustment, supporting the findings about Stone pine (*Pinus pinea* L.) seedlings under drought stress [67]. Soluble protein increased after 7 days of drought treatment, indicating that soluble protein plays a crucial role in the early adaptation of plants to drought. Our results are consistent with research into *Iris japonica*, which showed that Pro and protein enhanced cellular osmotic pressure and reduced water loss under drought stress [68]. The results showed that the soluble sugar, soluble protein, and Pro in triploid lines were higher than in diploid during the drought period, and fluctuations in the above three indicators were greater in diploid than in triploid, suggesting that diploid was more affected by stress. 

### 4.5. Antioxidant System of Triploid ‘Mantianxing’ under Drought

Water deficits caused by drought will lead to the excessive production and accumulation of reactive oxygen species (ROS), such as O_2_ and H_2_O_2_, which can result in injury to living tissues, macromolecules, and irrecoverable injuries within green tissues, even posing a risk of programmed cell death in plants [69]. Consequently, plants conduct various defense mechanisms to alleviate damage, and the antioxidants and secondary metabolites are the crucial strategies involved in ROS detoxification, enzyme/protein stabilization, and membrane protection, ultimately improving the drought tolerance of plants [70]. Different kinds of enzymes can effectively minimize the deleterious effects of ROS in different ways; therefore, antioxidant enzymes are considered crucial indicators for evaluating plant stress tolerance [71,72]. In studies on stress resistance in plants, polyploids showed a more efficient resistant system than diploids [73]. Moreover, the increased tolerance to environmental stress of polyploids was attributed to their higher antioxidant enzyme activity to a large extent [74,75]. As in most studies, we found that triploid *P. edulis* had a higher antioxidant enzyme activity to eliminate excess ROS, and similar enzymatic defense reactions were observed in the response of wheat species under drought stress [76]. Interestingly, the regulating mechanism of antioxidant enzyme activity in polyploid is an intricate process. Fakhrzad’s research into tetraploid wallflowers found that the extensive reprogramming of genes involved in hormonal signaling, reactive oxygen species detoxification, osmotic adjustment, and enhanced antioxidant defense system might contribute to the superior resistance of polyploid plants [77]. As for triploid ‘Mantianxing’, further studies are needed to understand the molecular regulation mechanisms behind drought resistance.

## 5. Conclusions

This present study evaluated and elucidated the differences in morphology, physiology, and biochemistry characteristics between triploid and diploid *P. edulis* ‘Mantianxing’ under drought stress. Research findings showed that triploid ‘Mantianxing’ effectively adjusted its photosynthetic parameters and maintained a higher chlorophyll content, *P*_n_, and chlorophyll fluorescence to respond to drought stress. Furthermore, triploid plants had a higher soluble sugar, soluble protein, and proline content to adjust the osmotic pressure of plant cells, as well as higher antioxidant enzymes to remove excessive ROS. Moreover, in this study, the changing trends in most indicator parameters in diploid plants were more significant than those of triploid, suggesting that diploids were more sensitive to drought stress. Our findings enhance our understanding of the response in polyploid *P. edulis* ‘Mantianxing’ to drought stress, provided an understanding of the response mechanisms in physiology and biochemistry of triploid under drought stress, as well as a significant implication for the development of the *P. edulis*-related industry in arid and semi-arid regions.

## Figures and Tables

**Figure 1 plants-13-01685-f001:**
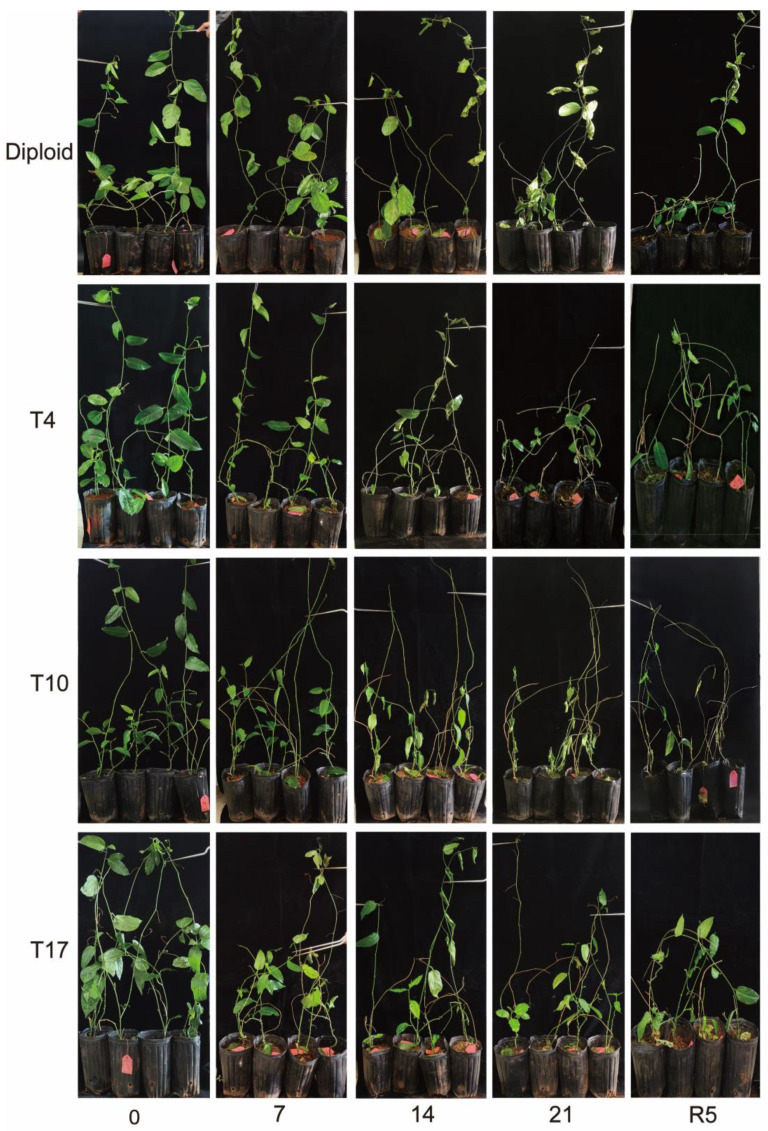
The morphology changes in diploid and triploid T4, T10, and T17 ‘Mantianxing’ under drought 0-day, 7 days, 14 days, 21 days, and rewatering 5 days (R5) after drought for 21 days.

**Figure 2 plants-13-01685-f002:**
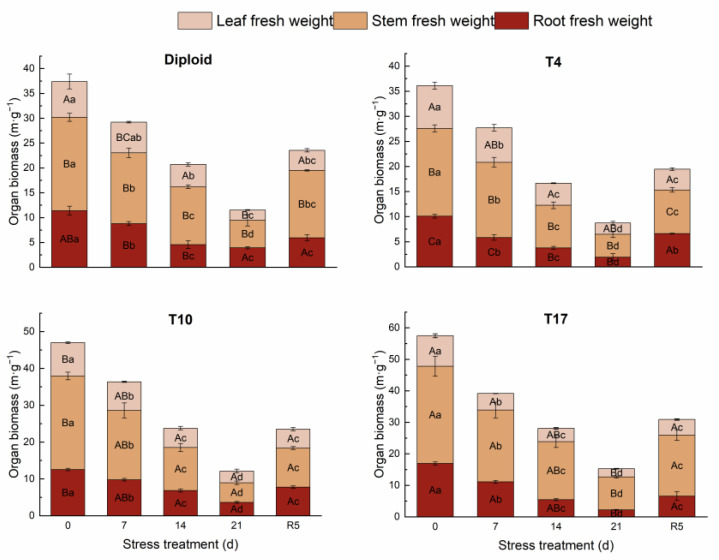
The changing trend of organ biomass of diploid and triploid T4, T10, and T17 ‘Mantianxing’ under drought 0-day, 7 days, 14 days, 21 days, and rewatering 5 days (R5) after drought for 21 days. Different capital letters mean differences between different ploidies in the same stress treatment, and different lowercase letters mean differences of the same ploidy in the different stress treatments. *p* < 0.05.

**Figure 3 plants-13-01685-f003:**
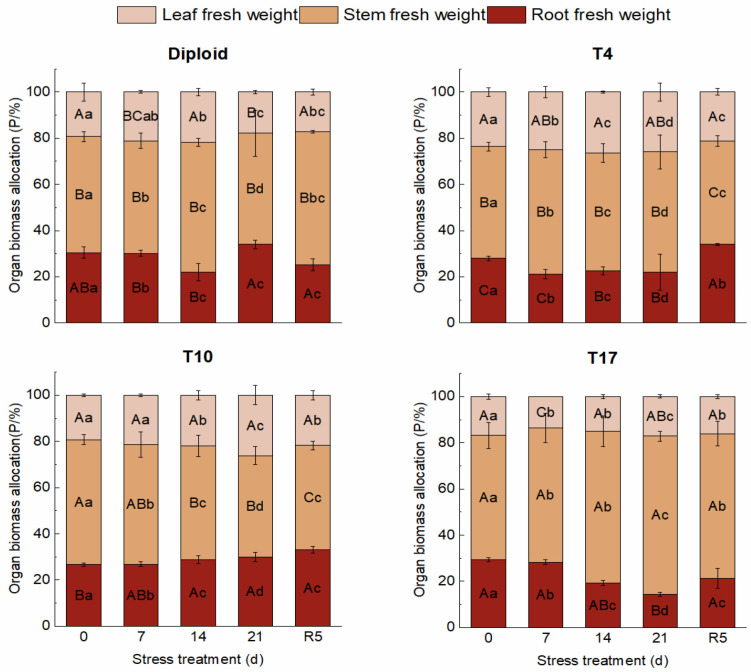
Organ biomass allocation of diploid and triploid T4, T10, and T17 ‘Mantianxing’ under drought 0-day, 7 days, 14 days, 21 days, and rewatering 5 days (R5) after drought for 21 days. Different capital letters mean differences between different ploidies in the same stress treatment, and different lowercase letters mean differences of the same ploidy in the different stress treatments. *p* < 0.05.

**Figure 4 plants-13-01685-f004:**
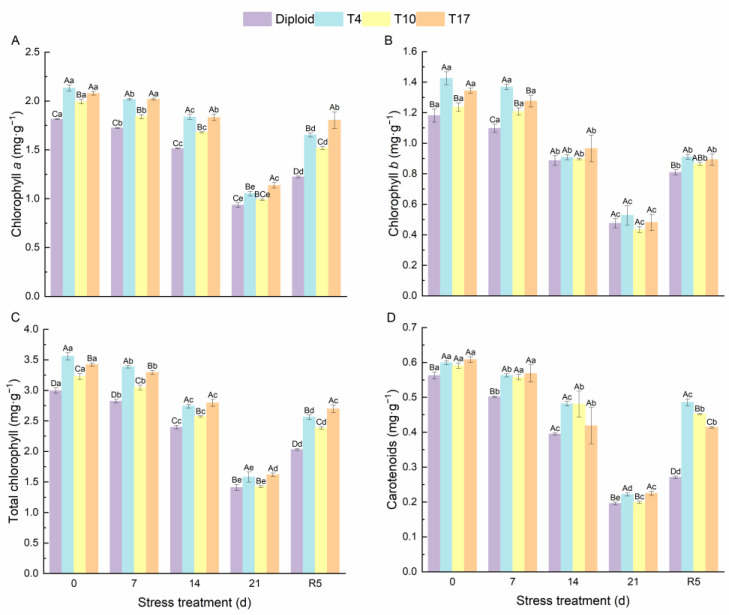
The content of chlorophyll *a* (**A**), chlorophyll *b* (**B**), total chlorophyll (**C**), and carotenoids (**D**) in diploid and triploid T4, T10, and T17 ‘Mantianxing’ under drought 0-day, 7 days, 14 days, 21 days, and rewatering 5 days (R5) after drought for 21 days. Different capital letters mean differences between different ploidy in the same stress treatment, and different lowercase letters mean differences of the same ploidy in the different stress treatments. *p* < 0.05.

**Figure 5 plants-13-01685-f005:**
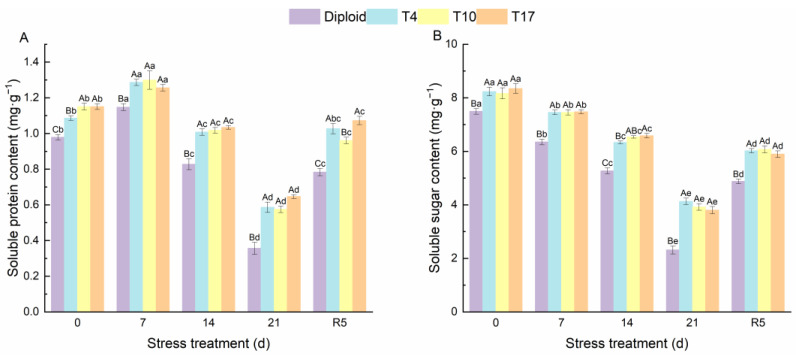
The soluble protein (**A**) and soluble sugar content (**B**) in diploid and triploid T4, T10, and T17 ‘Mantianxing’ under drought 0-day, 7 days, 14 days, 21 days, and rewatering 5 days (R5) after drought for 21 days. Different capital letters mean differences between different ploidy in the same stress treatment, and different lowercase letters mean differences of the same ploidy in the different stress treatments. *p* < 0.05.

**Figure 6 plants-13-01685-f006:**
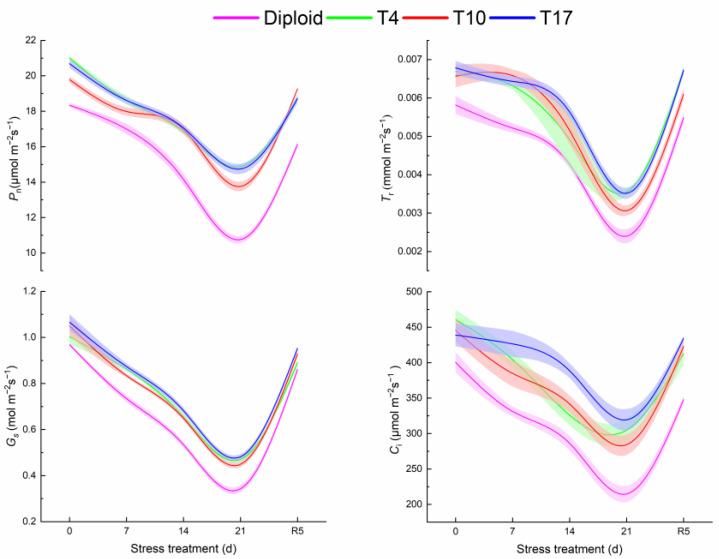
The response of photosynthetic system of diploid and triploid T4, T10, and T17 ‘Mantianxing’ under drought 0-day, 7 days, 14 days, 21 days, and rewatering 5 days (R5) after drought for 21 days: *P*_n_, net photosynthetic rate; *T*_r_, transpiration rate; *G*_s_, stomatal conductance; *C*_i_, intercellular CO_2_ concentration.

**Figure 7 plants-13-01685-f007:**
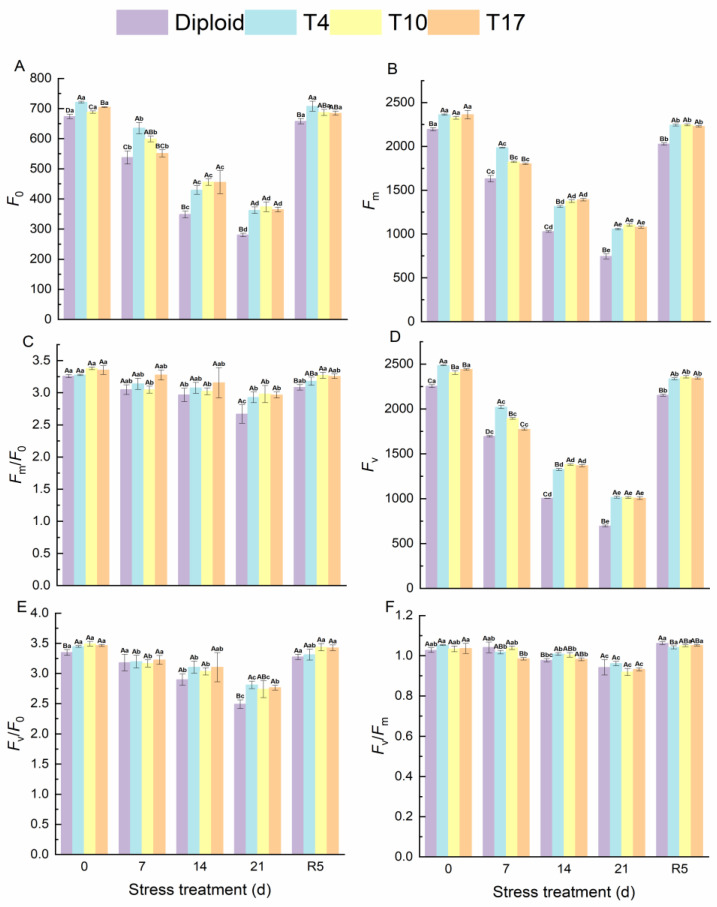
Chlorophyll fluorescence of diploid and triploid T4, T10, and T17 ‘Mantianxing’ under drought 0-day, 7 days, 14 days, 21 days, and rewatering 5 days (R5) after drought for 21 days. (**A**): minimum fluorescence yield in the absence of photosynthetic light (*F*_0_); (**B**): maximum fluorescence yield in the absence of photosynthetic light (*F*_m_); (**C**): electron transfer rate (*F*_m_/*F*_0_); (**D**): variable fluorescence (*F*_v_); (**E**): the ratio of photochemical to nonphotochemical processes (*F*_v_/*F*_0_); (**F**): the maximum quantum yield of PS II (*F*_v_/*F*_m_). Different capital letters mean differences between different ploidy in the same stress treatment, and different lowercase letters mean differences of the same ploidy in the different stress treatments. *p* < 0.05.

**Figure 8 plants-13-01685-f008:**
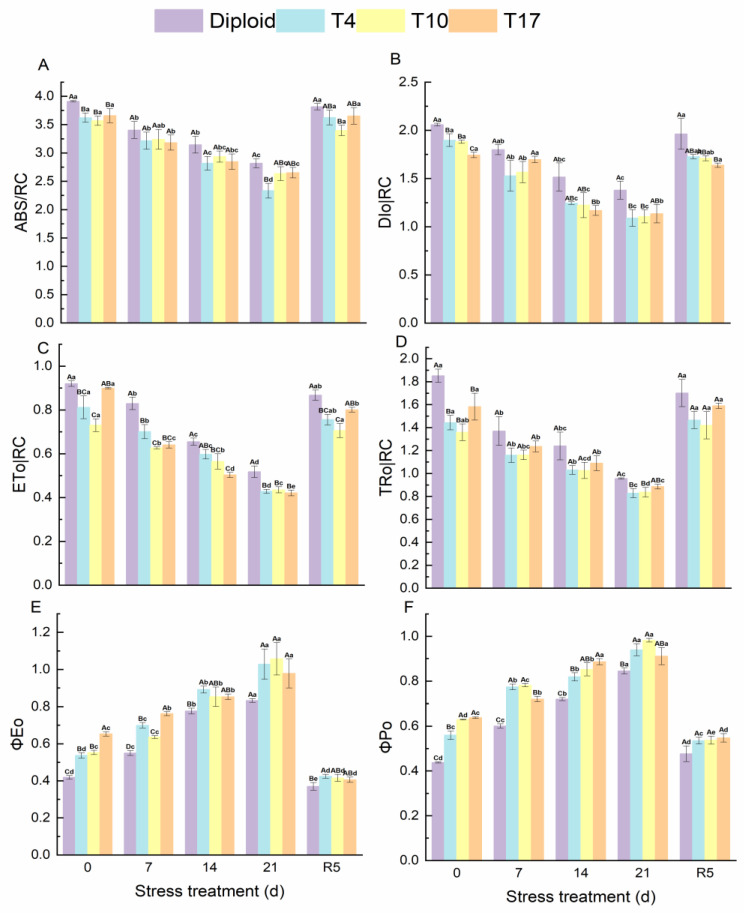
Energy flow and distribution in PS Ⅱ reaction center of diploid and triploid T4, T10, and T17 ‘Mantianxing’ under drought 0-day, 7 days, 14 days, 21 days, and rewatering 5 days (R5) after drought for 21 days. (**A**): light energy absorbed by a unit reaction center (ABS/RC); (**B**): dissipation at the level of the antenna chlorophylls (DIo/RC); (**C**): energy captured per unit reaction center for electron transfer (ETo/RC); (**D**): the specific energy fluxes (per reaction center, RC) for trapping (TRo/RC); (**E**): probability that an absorbed photon will move an electron into the electronic transport chain (ΦEo); (**F**): the PS Ⅱ maximum photochemical efficiency (ΦPo). Different capital letters mean differences between different ploidy in the same stress treatment, and different lowercase letters mean differences of the same ploidy in the different stress treatments. *p* < 0.05.

**Figure 9 plants-13-01685-f009:**
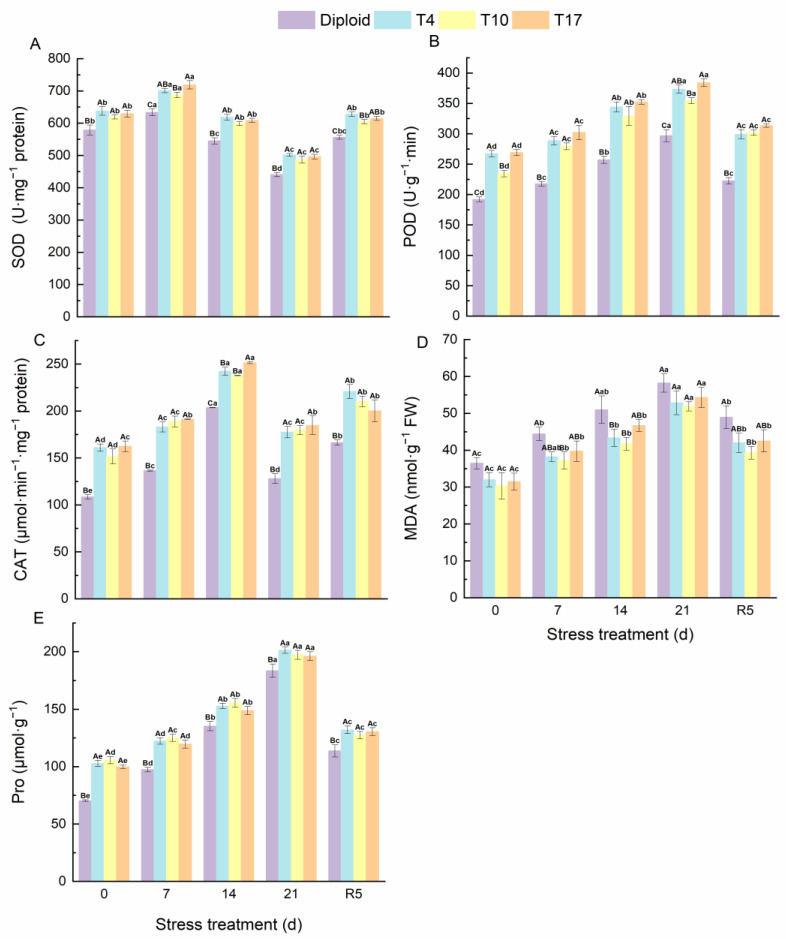
The antioxidant enzyme activity of diploid and triploid T4, T10, and T17 ‘Mantianxing’ under drought 0-day, 7 days, 14 days, 21 days, and rewatering 5 days (R5) after drought for 21 days: SOD, superoxide (**A**); POD, peroxidase (**B**); CAT, catalase (**C**); MDA, malonaldehyde (**D**); Pro, proline (**E**). Different capital letters mean differences between different ploidy in the same stress treatment, and different lowercase letters mean differences of the same ploidy in the different stress treatments. *p* < 0.05.

**Figure 10 plants-13-01685-f010:**
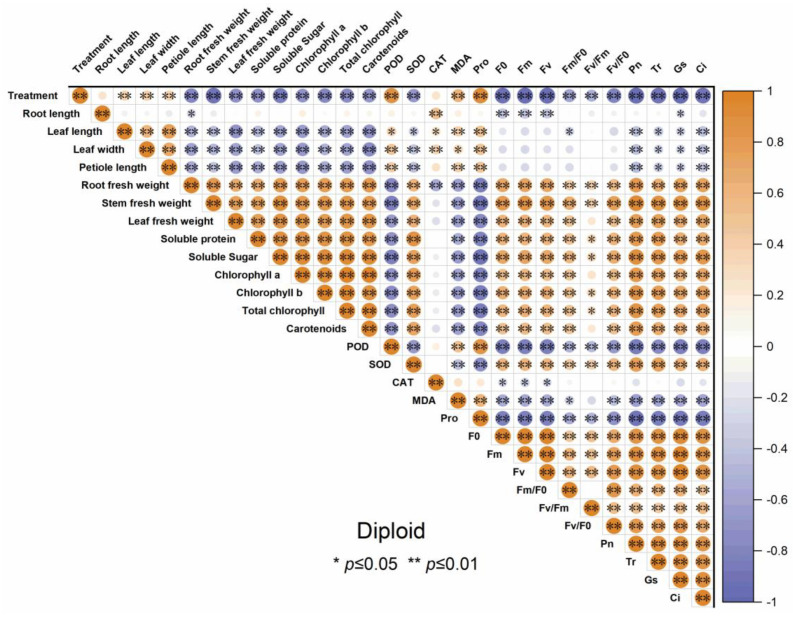
Correlation analysis of different indicators of diploid ‘Mantianxing’ under drought 0-day, 7 days, 14 days, 21 days, and rewatering 5 days (R5) after drought for 21 days. Note: SOD, superoxide; POD, peroxidase; CAT, catalase; MDA, malonaldehyde; Pro, proline; *P*_n_, net photosynthetic rate; *T*_r_, transpiration rate; *G*_s_, stomatal conductance; *C*_i_, intercellular CO_2_ concentration. The size of the circle represents the absolute value of the correlation coefficient.

**Figure 11 plants-13-01685-f011:**
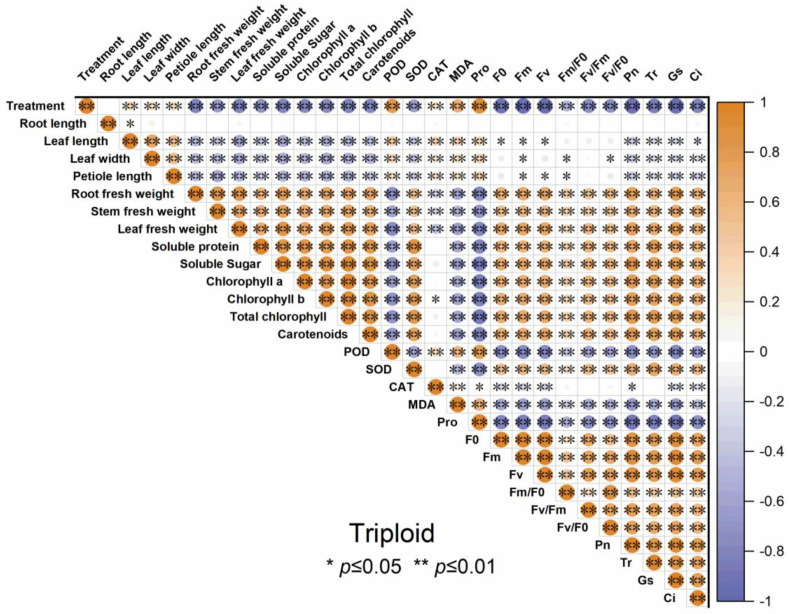
Correlation analysis of different indicators of triploid T4, T10, and T17 ‘Mantianxing’ under drought 0-day, 7 days, 14 days, 21 days, and rewatering 5 days (R5) after drought for 21 days. Note: SOD, superoxide; POD, peroxidase; CAT, catalase; MDA, malonaldehyde; Pro, proline; *P*_n_, net photosynthetic rate; *T*_r_; transpiration rate; *G*_s_, stomatal conductance; *C*_i_, intercellular CO_2_ concentration. The size of the circle represents the absolute value of the correlation coefficient.

**Table 1 plants-13-01685-t001:** Morphology traits of diploid and triploid T4, T10, and T17 ‘Mantianxing’ under drought 0-day, 7 days, 14 days, 21 days, and rewatering 5 days (R5) after drought for 21 days.

Indicator	Ploidy	0	7	14	21	R5
Root length (cm)	Diploid	636.32 ± 42.95 Bb	1450.41 ± 313.32 Aa	543.00 ± 50.72 Aab	507.55 ± 58.77 Ab	952.11 ± 307.17 Ab
	T4	823.78 ± 139.13 Aa	897.48 ± 145.42 Aa	769.77 ± 228.89 Aa	671.71 ± 195.53 Aa	822.73 ± 79.76 ABa
	T10	924.42 ± 98.28 Aa	1333.88 ± 677.14 Aa	819.12 ± 72.98 Aa	619.70 ± 24.68 Ba	924.42 ± 98.28 Aa
	T17	975.07 ± 315.70 Aa	1227.95 ± 273.99 Aa	886.43 ± 113.72 Aa	827.97 ± 116.06 Aa	1026.90 ± 287.79 Aa
Leaf length (cm)	Diploid	9.35 ± 0.21 Ba	8.63 ± 0.15 Cb	8.05 ± 0.19 Bb	7.36 ± 0.27 Bc	9.89 ± 0.23 Ba
	T4	9.47 ± 0.15 Bb	8.76 ± 0.11 BCc	8.24 ± 0.11 Bd	7.78 ± 0.11 ABe	10.07 ± 0.14 Ba
	T10	9.51 ± 0.23 Bb	9.18 ± 0.20 Bb	8.38 ± 0.24 Bc	7.86 ± 0.25 ABc	10.39 ± 0.38 Ba
	T17	10.48 ± 0.21 Ab	9.82 ± 0.23 Ab	9.06 ± 0.22 Ac	8.09 ± 0.20 Ad	11.58 ± 0.32 Aa
Leaf width (cm)	Diploid	6.47 ± 0.15 Aa	6.03 ± 0.12 Ab	5.20 ± 0.10 Ac	4.40 ± 0.18 Ad	6.57 ± 0.13 Aa
	T4	4.46 ± 0.15 Cb	4.16 ± 0.09 Cb	3.64 ± 0.15 Cc	3.42 ± 0.14 Ac	5.13 ± 0.18 Ba
	T10	5.30 ± 0.25 Bb	4.80 ± 0.23 Bbc	4.40 ± 0.14 Bc	3.58 ± 0.21 Ad	6.26 ± 0.26 Aa
	T17	5.38 ± 0.20 Ba	4.89 ± 0.21 Ba	4.20 ± 0.26 Ba	3.58 ± 0.09 Aa	6.08 ± 0.29 Aa

Note: Different capital letters mean differences between different ploidies in the same stress treatment, and different lowercase letters mean differences in the same ploidy in the different stress treatments. *p* < 0.05.

## Data Availability

Data are available upon reasonable request.

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
