# Peer review of "The Response and Evaluation of Morphology, Physiology, and Biochemistry Traits in Triploid Passiflora edulis Sims ‘Mantianxing’ to Drought Stress"

_plants, 2024, doi:10.3390/plants13121685_

Round 1

Reviewer 1 Report

Comments and Suggestions for Authors

General Comments:

I have reviewed the manuscript titled "The Response and Evaluation of Morphology, Physiology and Biochemistry Traits in Triploid Passiflora edulis sims ‘Mantianxing’ to Drought Stress written by Xin Su et al., intended for publication in Plants. Author primarily investigates various morphology, physiology, and biochemistry variations of triploid ‘Mantianxing’ under drought, our study measured drought-related indicators 16 under natural drought condition. Author mainly found that the drought stress significantly affected the growth and development of P. edulis ‘Mantianxing’. The wilting and the variations of biomass allocation may be the outcome of plant drought coping strategies.

The research is well-conducted and informative triploid P. edulis ‘Mantianxing’ showed higher Pn and chlorophyll fluorescence, osmotic adjustment substances, and antioxidant enzyme activity under drought stress as well as better performance after rewatering. However, there is a lack of cohesive storytelling and potential references, so mention the potential reference that I indicated below. Consequently, I recommend that the manuscript undergo "MAJOR REVISION."

 Major Suggestions

1)Abstract: The authors should further abstract by emphasizing the novelty of the results and little more expanding the methodology section. They should rephrase the results to make them clearer and more engaging for a wider audience. It's crucial to elucidate how this study benefits society.

2)Introduction: Author should mention more references for further clarity the text in each traits in introduction section. Drought generally involves morphological characteristics specially reduction of leaf size or alternation of leaf length and leaf width characteristics as well as reduction of photosynthesis and water movement for the plants DOI:10.1016/j.scienta.2018.11.021, this article well describe the traits related to the drought.

3)Novelty of the Study: Ensure the research objectives are clearly connected to the research hypothesis. Clearly state the hypotheses and research objectives, as they are essential for contextualizing the study. Without them, the text may lack clarity.

4)Conciseness: Remove unnecessary and less important text to streamline the manuscript because introduction and discussion section can be more concise and informative.

Line-by-Line Comments:

5)Line 84-100: Hypothesis and research objectives: Author well presented the objectives or main aim of the study in the Ln. 96-98. However, the research hypothesis is not still much clearly presented. Please mention the research hypothesis and well-connect these two parts in the single paragraph by rephrasing the last paragraph of the introduction. The hypothesis should be very clear because, without appropriate literature, questions, or hypotheses in the introduction section the entire text will be unclear.

6)Line 196 (Result; Fig.1): Author should mention all necessary text in the legend of each figure. Please expand the text of Figure 1 because it was not much clear. Each figure legend should describe the picture well. Accordingly, please improve the others (if lacking).

7) Line no. 330 (Discussion): Author should address why arises the antioxidant and secondary metabolites under drought? These articles better clarify (1) https://doi.org/10.1038/s41598-019-55889 (2) https://doi.org/10.1016/j.scitotenv.2021.146466 about to produces the ROS when the plant exposed to the stress condition and plant produce antioxidant, flavonoids, and secondary metabolites.

8) Line no. 461 (Conclusion): I did not see the conclusion section separately made by author. Please remember that conclusion should not be repetitive in the abstract or a summary of the results section. I would love to read striking points and take-home messages that will linger in the readers’ minds. What is the novelty, how does the study elucidate some questions in this field, and the contributions the paper may offer to the scientific community?

9) Line no. 492 (References): please include more related citation, check their pattern and writing style, spell check, and other grammatical errors. moreover, the author should cut the old and less matching literature and include the latest literature some of them are above.

Good Luck!

Reviewer 2 Report

Comments and Suggestions for Authors

The paper is about The response and evaluation of morphology, physiology and biochemistry traits in triploid Passiflora edulis sims 'Mantianxing' to drought stress. Although the effect of ploidy in plants exposed to drought has already been studied, which takes away the originality of the paper, I consider that it is well done. The results have been well discussed and the techniques used are appropriate to explore the established objectives. 

Minor comments:  

-line 203 is triploid instead of tetraploid.

-Figure 10 is very small which does not make it easy to read.

-Change the pastel colors of the figures. 

Round 2

Reviewer 1 Report

Comments and Suggestions for Authors

Dear Author

I have read the revised manuscript. Entitled: The response and evaluation of morphology, physiology and biochemistry traits in triploid Passiflora edulissims ‘Mantianxing’ to drought stress in Plants MDPI. This is the second submission made by the author. The author addressed all the questions and suggestions that I raised the issue in the review. I satisfy the Author improved the abstract significantly. Author significantly improved their research hypothesis and well connected with the research objectives in this time. This manuscript improved the flow of writing, which was comparatively shallow in the original version but in this revised copy author very well addressed all the quarries and suggestions. Before accepting this manuscript, please check again the referencing. Further if there is anything needed to be revised by the author, especially English grammar, or spell check, I request this manuscript is currently in “Minor Revision” and the author may correct any further grammatical errors (if any) the author may improve in this stage.

Thank you.